# Global landscape of phenazine biosynthesis and biodegradation reveals species-specific colonization patterns in agricultural soils and crop microbiomes

Daniel Dar[1,2], Linda S Thomashow[3], David M Weller[3], Dianne K Newman[1,2]*

[1]Division of Geological and Planetary Sciences, California Institute of Technology, Pasadena, United States; [2]Division of Biology and Biological Engineering, California Institute of Technology, Pasadena, United States; [3]Wheat Health, Genetics and Quality Research Unit, USDA Agricultural Research Service, Pullman, United States

**Abstract** Phenazines are natural bacterial antibiotics that can protect crops from disease. However, for most crops it is unknown which producers and specific phenazines are ecologically relevant, and whether phenazine biodegradation can counter their effects. To better understand their ecology, we developed and environmentally-validated a quantitative metagenomic approach to mine for phenazine biosynthesis and biodegradation genes, applying it to >800 soil and plant-associated shotgun-metagenomes. We discover novel producer-crop associations and demonstrate that phenazine biosynthesis is prevalent across habitats and preferentially enriched in rhizospheres, whereas biodegrading bacteria are rare. We validate an association between maize and *Dyella japonica*, a putative producer abundant in crop microbiomes. *D. japonica* upregulates phenazine biosynthesis during phosphate limitation and robustly colonizes maize seedling roots. This work provides a global picture of phenazines in natural environments and highlights plant-microbe associations of agricultural potential. Our metagenomic approach may be extended to other metabolites and functional traits in diverse ecosystems.

**\*For correspondence:**
dkn@caltech.edu

## Introduction

Phenazines are heterotricyclic N-containing metabolites mainly produced by bacteria in soil and plant-root microbiomes (*Biessy and Filion, 2018*; *Mavrodi et al., 2010*), though some are also important in chronic human infections (*Sismaet et al., 2016*; *Wilson et al., 1988*). Members of this diverse family of redox-active metabolites act as broad-spectrum antibiotics by generating toxic reactive oxygen species (ROS) as well as interfering with cellular respiration chains (*Baron et al., 1989*; *Perry and Newman, 2019*). In addition to their activity as antibiotics, phenazines can benefit their producers in other ways: promoting survival under anoxic conditions (*Glasser et al., 2014*; *Wang et al., 2010*), facilitating biofilm development (*Dietrich et al., 2013*) and iron acquisition (*Wang et al., 2011*), and increasing the fitness of phenazine-producing (*phz*+) fluorescent pseudomonads in the plant rhizosphere (*LeTourneau et al., 2018*; *Mazzola et al., 1992*).

In the context of the plant microbiome, phenazines produced by fluorescent pseudomonads have been shown to protect major crops such as wheat (*Chen et al., 2018*; *Thomashow et al., 1990*) and tomato (*Chin-A-Woeng et al., 1998*) from disease-causing fungi and water molds. Moreover, naturally produced phenazine 1-carboxylic acid (PCA) measured in commercial wheat rhizospheres reached levels as high as 1 µg PCA per gram fresh roots, and strongly correlated with the number of cultured *phz*+ pseudomonads (*Mavrodi et al., 2012*). Nonetheless, the effective concentration of phenazines is likely to be considerably greater within soil microenvironments. Interestingly, the

pathogenic fungus, *Fusarium oxysporum* f. sp. radicis-lycopersici, was previously shown to colonize similar tomato root zones as plant-growth promoting *phz*⁺ pseudomonads, where they likely compete for plant-root exudates (*Bolwerk et al., 2003*). This evidence for 'right place at the right time' niche overlap provides a potential hint as to how phenazines, and possibly other natural antibiotics, can limit pathogen proliferation, even at low producer abundances and in large and complex environments such as plant roots. Phenazine-producing bacteria are positively correlated with warmer and more arid soils and lower species diversity (*Maestre et al., 2015*; *Mavrodi et al., 2012*), suggesting that in a warming climate, phenazine production may impact the composition of the microbial communities associated with certain crops. However, phenazines can also be degraded and consumed by other soil bacteria (*Costa et al., 2015*; *Yang et al., 2007*). The balance of phenazine production and degradation will ultimately dictate the extent to which phenazines will have agricultural relevance, yet the extent to which these processes are potentially important in agricultural environments is currently unknown.

Since their discovery and characterization in pseudomonads, genomic signatures of phenazine biosynthesis have been detected in a large set of phylogenetically diverse bacteria (*Hadjithomas et al., 2015*; *Mavrodi et al., 2010*). Despite this diversity, all producer genomes contain a conserved set of core biosynthesis genes *phzA/BCDEFG* (*Biessy and Filion, 2018*; *Mavrodi et al., 2010*; *Figure 1A*). Together, these proteins synthesize one of two main phenazines that act as precursors for all other derivatives: PCA in pseudomonads and phenazine-1,6-dicarboxylic acid (PDC) in most other species (*Figure 1B*). Phenazine chemical diversity is determined by specific auxiliary genes, which modify and decorate the core phenazine structure. These modifications can strongly affect the toxicity spectrum and function of different phenazine species (*Laursen and Nielsen, 2004*).

While the majority of phenazine research has focused on pseudomonads, the growing phylogenetic diversity of phenazine producers and metabolite variants that can be identified genomically suggests much remains to be learned. Importantly, the ecology of these metabolites in natural and agricultural environments is poorly understood. Furthermore, it is generally unknown which bacterial clades and phenazine derivatives are naturally associated with various major crops. While current culturing-based approaches work well for readily-culturable bacteria such as fluorescent pseudomonads, their ability to select for organisms representing the full phylogenetic diversity of phenazine producers is limited. Therefore, the development of quantitative and culture-independent methods is necessary to facilitate the discovery of novel plant-microbe associations between crops and phenazine producers.

Here, we combine computational metagenomics with field and targeted laboratory experiments to quantitatively capture the global landscape of phenazine antibiotic biosynthesis and biodegradation in soil and plant-associated environments, with the aim of uncovering novel agriculturally promising, plant-microbe associations.

## Results

### Constructing a comprehensive phenazine biosynthesis reference database

The accumulation of shotgun-metagenomic sequencing data in public repositories over the last decade has provided a rich database from which to perform global analysis of phenazine bioproduction in soil and crop environments. Importantly, metagenomic data are particularly suitable for mining new associations between novel phenazine producers and major crops.

Because all known phenazine producers contain the core phenazine biosynthesis genes, *phzA/BCDEFG*, almost always in single-copy, we reasoned that given a sufficiently broad database of these proteins, we could accurately identify and enumerate the levels of *phz*⁺ bacteria via shotgun-metagenomics (*Figure 1A*). We constructed a comprehensive *phzA/BCDEFG* reference gene set by mining the Integrated Microbial Genomes (IMG), IMG-ABC biosynthetic gene cluster atlas (*Hadjithomas et al., 2015*), as well as by conducting iterative homology searches. During this analysis, we noticed that highly similar copies of the *phzC* gene were common in non-producer genomes and therefore removed this gene from our analysis. In total, we assembled a list of 181 phenazine producer genomes (*Supplementary file 1*). The final reference set contained 100 proteobacterial

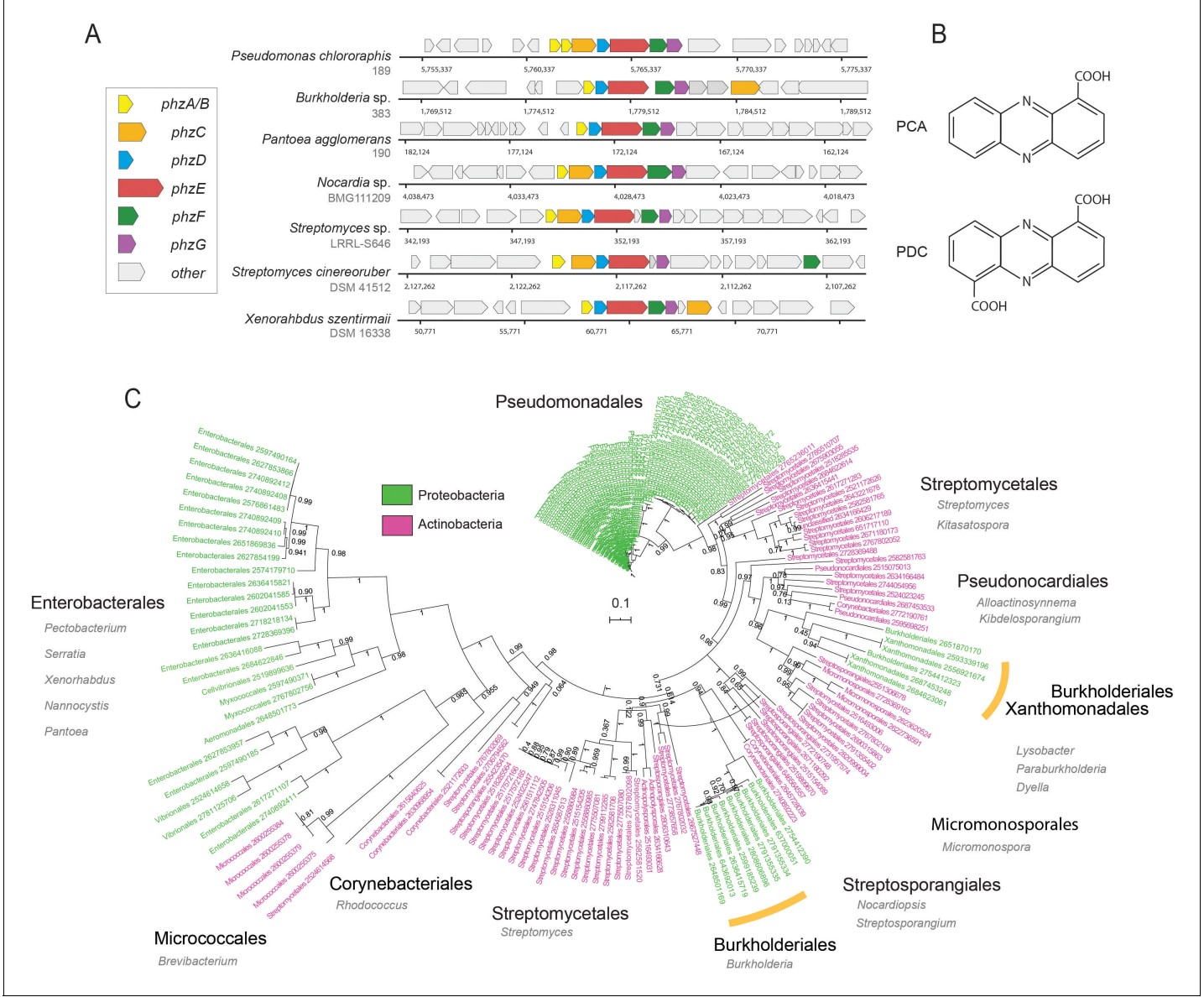

**Figure 1.** Diversity and phylogeny of phenazine producing bacteria assembled in this study. (**A**) Genomic regions containing the core phenazine biosynthesis genes in diverse bacteria. Core biosynthesis genes (*phzA/BCDEFG*) are colored as shown in the legend, all other surrounding genes are depicted in gray. (**B**) Chemical structures of the main phenazine precursors, PCA and PDC. (**C**) Phylogenetic tree constructed using the concatenated phenazine-biosynthesis protein sequences. Phylum level is color coded according to the legend and specific clades are annotated around the tree (orders in black and genera in gray). Orange bars highlight Xanthomonadales and Burkholderiales species. Tree scale and bootstrap values are depicted in the figure.

and 81 actinobacterial genomes representing a total of 15 phylogenetic Orders, exemplifying the diversity of phenazine producers (*Figure 1C*; *Supplementary file 1*).

To explore the evolutionary relationships between different *phz*+ bacteria, we constructed a phylogenetic tree using the above concatenated core-biosynthesis sequences. Notably, species organization in this tree substantially differ from the known phylogenetic relationship of *phz*+ bacteria in our database. While *Pseudomonas* species are highly clustered, other clades form surprising groupings. For example, phenazine-biosynthesis actinobacterial sequences of the Micrococcales order (e. g., *Brevibacterium linens*) are more similar to those from proteobacteria of the Enterobacterales, Myxococcales and Vibrionales orders (*Figure 1C*). In addition, Burkholderiales and Xanthomonadales are both embedded deep within the actinobacterial part of the tree rather than near other

proteobacteria (*Figure 1C*). These results support and expand on previous smaller scale analyses made with the *phzF* gene (*Mavrodi et al., 2010*), and suggest that multiple horizontal gene transfer (HGT) events have likely shaped the evolution of phenazine biosynthesis.

## A quantitative approach for measuring phz⁺ bacteria levels via shotgun-metagenomics

While metagenomic sequencing reads can be translated and mapped against the protein database described above, extracting quantitative and ecologically intuitive information about phenazine producer abundance is not trivial. One major complication is the uncertainty regarding the composition of the background bacterial community as well as other confounding factors that vary among environments. For example, different environments may contain different bacterial titers and some samples, especially those extracted from eukaryotic hosts, can contain high and variable amounts of non-bacterial DNA (host DNA, sequencing artifacts, other microbes such as Fungi, Protists, Archaea etc.). This variability in 'effective coverage' confounds normalization by library coverage and thus limits comparison between samples. This issue can be overcome by enumerating the total number of reads that map to bacteria. However, most environments contain substantial amounts of bacterial DNA for which no reference genome exists.

Employing universal marker genes as a reference point for normalizing gene abundance was previously found to be effective in mitigating these potential issues (*Manor and Borenstein, 2015*). We therefore generated a second reference set representing 'total-bacteria' that is based on the bac120 genes previously used in phylogenetic classification (*Parks et al., 2018*). We selected the top 25 most ubiquitous single-copy genes in the bac120 set, all of which are estimated to appear in >92.9% of known genomes (*Supplementary file 2*). For each such reference gene we acquired ~25,000 protein sequences from a comprehensive set of dereplicated genomes and assembled metagenomes (*Parks et al., 2018*). Together, these reference genes distill a metagenomic signal representing 'total-bacteria' in a manner that is robust to variation in community composition and stochastic parameters such as host DNA contamination.

We devised a computational procedure for estimating the percentage of *phz*⁺ bacteria out of all bacteria in a microbiome of interest (*Figure 2A*). In this approach, translated metagenomic sequencing reads are first mapped to a merged protein database that contains both phenazine biosynthesis and the 'total-bacteria' reference genes. For every gene (phenazine or reference) total read counts are normalized by the average gene size. A 'total-bacteria' score is then calculated by taking the median, size-normalized signal across all 25 reference genes. A similar median score is calculated across the six phenazine biosynthesis genes (merging *phzA* and *phzB*, which share highly similar sequences). Using the median in this analysis rather than the average provides robustness to non-specific signal (for example, cases where only one or two of the genes are highly abundant). The fraction of *phz*⁺ bacteria is calculated by dividing the phenazine-specific score by the total-bacteria score (*Figure 2A*).

We validated this approach in silico by generating synthetic metagenomes constructed using phylogenetically diverse bacteria (*Supplementary file 3*). We then tested different library coverages (5, 10, 15 and 20 million reads), producer abundances (0–5% total *phz*⁺ bacteria) and common *phz*⁺ taxonomic groups (*Pseudomonas* and/or *Streptomyces*). For each condition (e.g., 1% *Pseudomonas phz*⁺ with a total library of 10 million reads) we conducted a dozen technical replicates, resulting a total of 1296 simulated communities. In addition, for each such technical replicate, the abundance of each non-producer species was randomly re-generated each time to better mimic natural variations.

Because environmental samples may contain *phz*⁺ bacteria with *phzA/BDEFG* protein sequences that are somewhat different from those in our reference set, we first tested the specificity of our approach at increasingly loose mapping stringencies (*Figure 2—figure supplement 1*). Overall, we find strong and agreement between the 'ground-truth' producer levels and the results obtained using our approach, even at a mapping threshold of 80% amino-acid identity. These results indicate that this method has the potential to accommodate new genomes with substantial sequence variation, while maintaining a high degree of accuracy (*Figure 2B*; *Figure 2—figure supplement 1*; R = 0.99; Pearson). Importantly, our simulations capture the total *phz*⁺ bacteria level even in communities that contain more than one producer (*Figure 2B*) and the relative proportions of these different *phz*⁺ bacteria are measured with low error (*Figure 2C–D*). In addition, our approach could identify as little as 0.1% *phz*⁺ bacteria, even in relatively low coverage metagenomic samples (5–

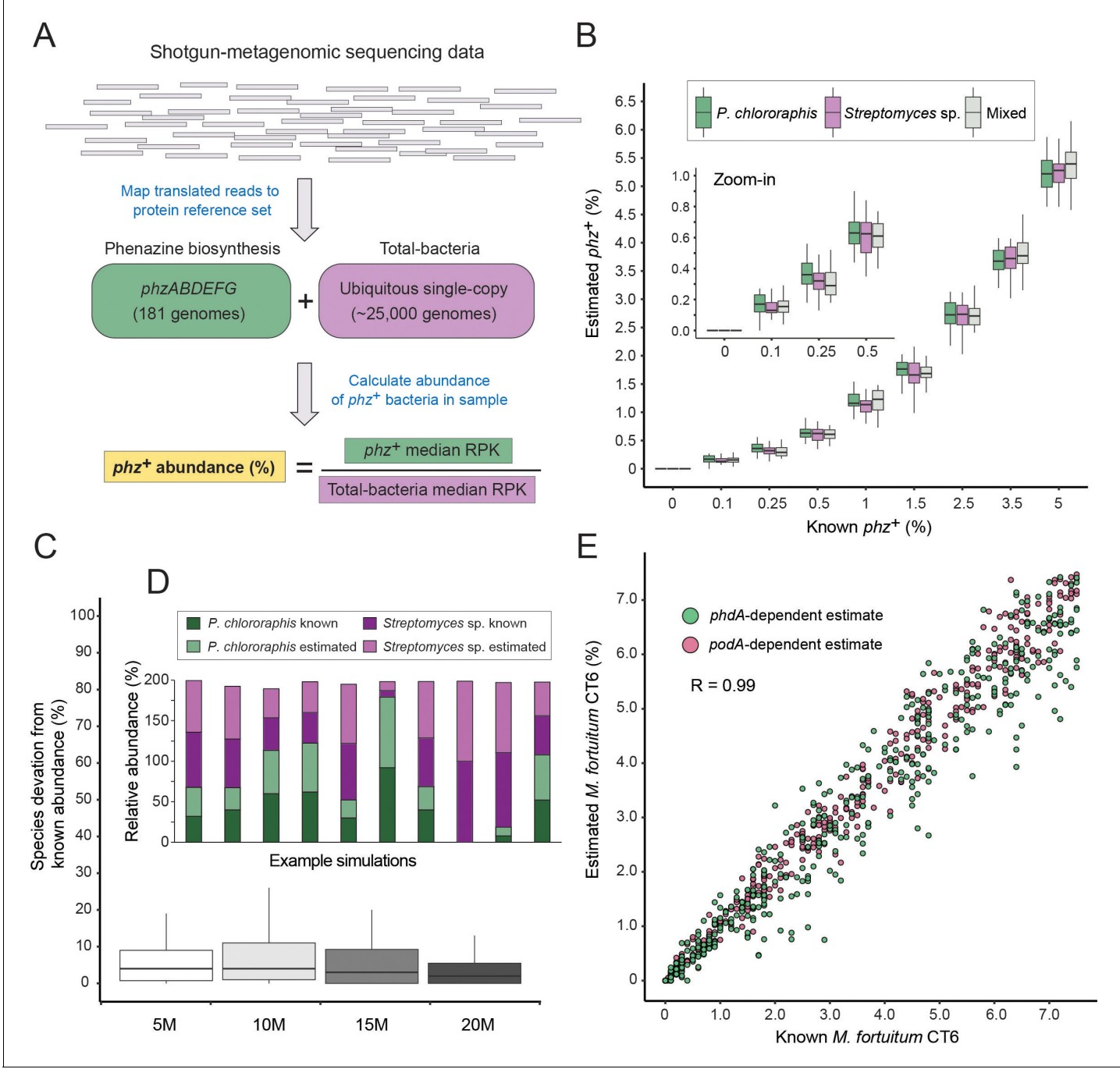

**Figure 2.** A method for estimating the abundance of phz+ bacteria using shotgun-metagenomics. (**A**) Schematic representation of the computational approach. Phenazine producer abundance is calculated using the median Reads Per Kilobase (RPK) levels. (**B**) Validation results using simulated communities. Box plots represent different producer combinations as indicated in the legend. Each boxplot represents data from simulations performed with 80% amino-acid identity threshold and across 5–20M library coverages (n = 48 per individual boxplot). The black bar represents the box median. (**C**) Accuracy in measuring relative abundance of multiple $phz^+$ species in a mixed community. Y-axis shows the difference between known and estimated species abundance across the indicated library coverages, considering samples with >0.1% $phz^+$ bacteria. (**D**) Example of individual simulation results with two different $phz^+$ bacteria. Similar heights of dark and light column portions represent good agreement between known and estimated levels. (**E**) Scatter plot depicting phenazine degrader (*M. fortuitum* CT6) frequency estimates at a gradient of known levels in simulated metagenomes. *M. fortuitum* levels are estimated using either *phdA* or *podA* genes, shown in pink and green, respectively.

The online version of this article includes the following figure supplement(s) for figure 2:

**Figure supplement 1.** Benchmarking with mapping and library coverage parameters.

**Figure supplement 2.** Low variation in bac120 reference gene coverage across samples.

10M reads) and detected no producers in any of the negative controls (*Figure 2B*; *Figure 2—figure supplement 1*).

In principle this approach should be readily generalizable to study other genes and functions beyond phenazine biosynthesis. By way of illustration, we assembled a third reference set containing the phenazine degradation genes: *phdA* (*Costa et al., 2018*) and *podA* (*Costa et al., 2017*) that enable mycobacteria to consume phenazines as a carbon source. We find that this approach accurately captures the levels of *phz+* and phenazine-degrading bacteria (*Mycobacterium fortuitum* CT6) in the same simulated communities and across a wide range of *phdA+*/*podA+* bacterial abundances, and library coverages (*Figure 2E*).

## Field validation in the wheat rhizosphere

While synthetic metagenomic data are essential for quantitative evaluation purposes, environmental samples are inherently more complex. We therefore sought to further test our approach under relevant field conditions. Wheat grown in the Inland Pacific Northwest, USA, is regularly colonized by *phz+* fluorescent pseudomonads that can protect these crops from disease-causing fungi (*Mavrodi et al., 2018*; *Mavrodi et al., 2013*). Thus, this environment provides an optimal test bed, where our metagenomic analysis can be directly compared with targeted culturing results, and where the levels of *phz+* bacteria potentially represent agriculturally meaningful levels.

We sampled 6-weeks-old wheat seedlings grown in a field at the Washington State University's Lind Agricultural Research Station in the Inland Pacific Northwest. As a reference, bulk field soil was collected between planted wheat rows (*Figure 3A*; *Figure 3—figure supplement 1*). The number of *phz+* pseudomonads was determined using a culturing-based approach and was estimated at $6.5 \times 10^6$ cfu/g roots in the wheat rhizosphere and at $1.6 \times 10^3$ cfu/g in bulk soil, in agreement with previous reports (*Mavrodi et al., 2018*). Metagenomic DNA was extracted from seedling rhizospheres (n = 5) and bulk field soil (n = 3) and shotgun-sequenced, producing 100 bp single-end reads. The resulting metagenomic sequences were then analyzed using the computational approach as described above (*Figure 2A*). In agreement with the culturing results, we find clear signal representing *phz+* bacteria in these samples, with an average relative abundance of 1.1% (+ / - 0.4%) in the wheat seedling rhizosphere and 0.32% (+ / - 0.14%) of the bulk field soil microbiome (*Figure 3A*; *Supplementary file 4*).

Surprisingly, a closer examination of the *phz+* bacteria in our rhizosphere metagenomes showed that two different bacterial clades occur at roughly the same abundance in this niche: the expected, culture-validated fluorescent pseudomonads, and *Streptomyces* spp., which were not previously known to occur in meaningful levels in this environment (*Figure 3B*). The average *phz+* pseudomonad level is thus estimated by our approach at ~0.4% of the total rhizosphere bacteria. While the enrichment-culture and metagenomic approaches are not directly comparable, they can be used together to extrapolate the number of bacteria in the rhizosphere. Indeed, combining the average *phz+* pseudomonad cfu/g with their average estimated percent out of all bacteria suggests a wheat seedling rhizosphere bacterial load of ~$10^9$ bacteria per gram roots, in agreement with other rhizosphere load estimates (*Bakker et al., 2013*; *Mavrodi et al., 2018*). Bulk field soil *phz+* bacteria were mainly composed of *Streptomyces* and other actinobacteria of different taxonomies that were found at a lower abundance (*Figure 3B*; *Supplementary file 4*).

Notably, three species in our reference set comprise >87% of the *phz+ Pseudomonas* rhizosphere sequencing reads (*Figure 3C*). These reads mapped to multiple strains, some of which were more highly represented, including, *P. fluorescens* (recently renamed *P. synxantha*) LBUM223, *P. chlororaphis aureofaciens* 30–84 and *P. orientalis* DSM 17489. Whereas these *P. fluorescens* and *P. orientalis* genomes contain a solo *phzA/BCDEFG* operon, *Pseudomonas chlororaphis aureofaciens 30-84* also contains an adjacent *phzO* modification gene, suggesting that pseudomonads in this environment produce PCA and 2-hydroxyphenazine-1-carboxylic acid (2-OH-PCA) (*Delaney et al., 2001*). Overall, these data are consistent with sequenced *Pseudomonas* previously isolates from similar environments (*Biessy et al., 2019*; *Parejko et al., 2012*).

The *Streptomyces* mapped reads in our data mapped to many species rather than just a few genomes, as seen for *Pseudomonas* (*Figure 3C*). Nevertheless, binning these *Streptomyces* genomes according to similarity in their phenazine genomic regions revealed that three main groups occupy the wheat rhizosphere (*Figure 3D–E*). The first, composed of only a single genome in our database (*Streptomyces* sp. URHA0041) contains two putative modification genes: a *phzM*

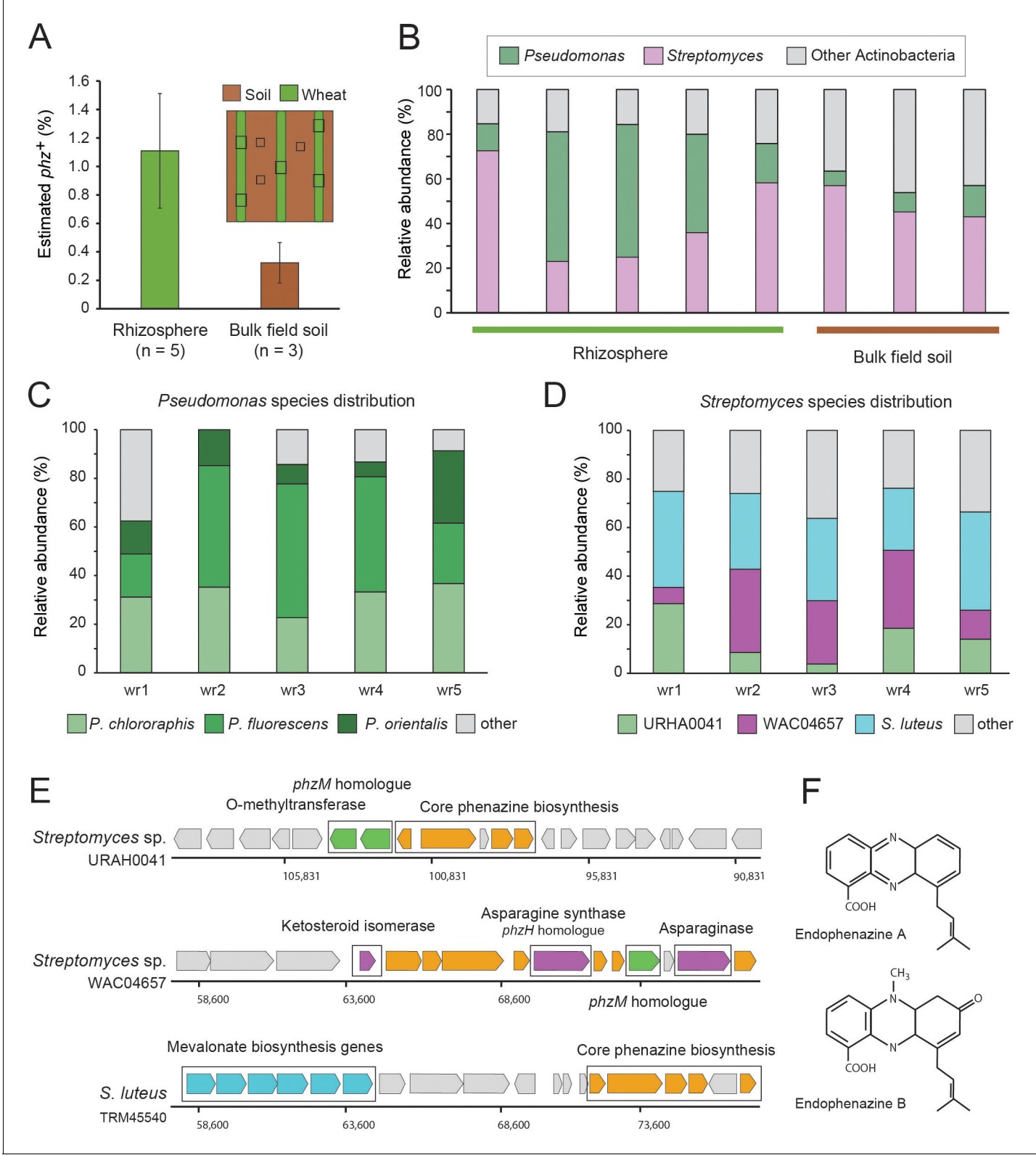

**Figure 3.** Field validation in the wheat rhizosphere. (**A**) Illustration of sample collection and metagenomic results. Green and brown stripes represent planted wheat rows and field soil, respectively. Black boxes depict sampling sites. Estimated *phz*+ levels in wheat rhizospheres and soil collected between planted rows. Error bars represent standard deviation. (**B**) Order level taxonomic distribution of *phz*+ bacteria within each sample. Y-axis describes the relative abundance of each order out of all *phz*+ bacteria. (**C–D**) Species level *phz*+ distributions across wheat rhizosphere samples (wr1-

*Figure 3 continued on next page*

*Figure 3 continued*

wr5). (**E**) Genomic regions displaying phenazine core and modification genes in the main *Streptomyces* groups detected in the rhizosphere. Specific modification genes are annotated in the figure. (**F**) Chemical structures of representative prenylated phenazines predicted to be produced by the mevalonate pathway highlighted in the *S. luteus* genomic region in E.

The online version of this article includes the following figure supplement(s) for figure 3:

**Figure supplement 1.** Wheat field sampling site.

homologue (~55% identity), whose product makes the precursor metabolite (5-Me-PCA) in the phenazine pyocyanin, in *P. aeruginosa* (*Mavrodi et al., 2001*), and a predicted O-methyltransferase (*Figure 3D–E*). The second group, represented by *Streptomyces* sp. WAC04657, contained four putative modification genes: the *phzM* homologue, a protein annotated as ketosteroid isomerase, an asparagine synthase, homologous to the *Pseudomonas phzH* gene (42% identity) that converts PCA to phenazine-1-carboxamide (PCN) (*Chin-A-Woeng et al., 2001*), and an asparaginase of unknown function (*Figure 3D–E*). The third and most abundant group, represented by *S. luteus* TRM45540, contained a conserved mevalonate pathway operon previously shown to produce prenylated phenazines called endophenazines (*Laursen and Nielsen, 2004*; *Saleh et al., 2009*; *Figure 3D–F*). These data imply that in contrast to pseudomonads, the *phz*⁺ *Streptomyces* community is likely to produce multiple chemically diverse phenazines in this environment, some of which await description.

## Global biogeographical analysis of phenazine biosynthesis and biodegradation in soil and plant-associated microbiomes

Our metagenomic-based approach provides a means to describe in quantitative terms which phenazine producers and consumers exist in a habitat of interest. To acquire a more complete picture of phenazine biosynthetic and catabolic potentials in soils and crop rhizospheres, we applied our approach to an additional 799 publicly available shotgun-metagenomes (*Supplementary file 4*; Methods). These new samples were combined with the eight wheat and field soil samples described above and were broadly classified as either soil (58.7%; 474/807) or rhizosphere (41.3%; 333/807). Bulk soil samples were further classified into sub-categories according to the environment from which they were collected (e.g., agricultural field, forest, grassland etc.). Similarly, rhizosphere samples were sub-classified according to the plant species from which they were extracted (*Figure 4—figure supplement 1*; *Supplementary file 4*).

Estimated $phz^+$ levels range from 0% to 2.7% of total environmental bacteria, with >28% (229/807) of all samples containing at least 0.25% $phz^+$ bacteria in their microbiome (*Figure 4A*; *Supplementary file 4*). Notably, we find a 1.9-fold higher average $phz^+$ level in rhizospheres compared to bulk soils (*Figure 4A*; p=$4.2\times10^{-17}$; Wilcoxon). Furthermore, samples with high total producer levels (at least 0.5% $phz^+$) are 8.9-fold enriched in samples annotated as rhizosphere compared to bulk soil. Higher $phz^+$ levels are particularly common in crops such as sugarcane, wheat, barley, maize, and citrus, as well as the model plant *Arabidopsis thaliana* (*Figure 4A*; *Supplementary file 4*). Lower yet significant $phz^+$ levels are common in diverse agricultural field soils (rice paddies, soybean/peanut fields, etc.) as well as in grasslands and forest soils (*Figure 4A*; *Supplementary file 4*). Together, these results outline the global distribution of phenazine biosynthetic potential across diverse soil and crop-associated habitats and reveal a trend of enrichment for these metabolites in plant-root microbiomes (*Figure 4A–B*; *Supplementary file 4*).

Next, we examined which phenazine producer clades occurred at significant levels in different habitats. To this end, we calculated the relative abundance of major taxonomic groups in individual samples and more generally across all samples belonging to the same habitat (averaged associations). Strikingly, we find that $phz^+$ *Streptomyces* represent a major portion of $phz^+$ bacteria in most habitats (*Figure 4B*). Furthermore, $phz^+$ *Streptomyces* are detected in substantial levels in over 67% (154/229) of $phz^+$ rich samples (containing at least 0.25% $phz^+$ bacteria), both soil and rhizosphere derived (*Figure 4—figure supplement 2*). In almost all cases, $phz^+$ *Streptomyces* can be found along with other producers from different clades, including various actinobacteria and proteobacteria (*Figure 4—figure supplement 2*; *Supplementary file 4*). Higher-resolution analysis of the mapped phenazine biosynthesis genes suggests *Streptomyces* sp. WAC04657 is a cosmopolitan

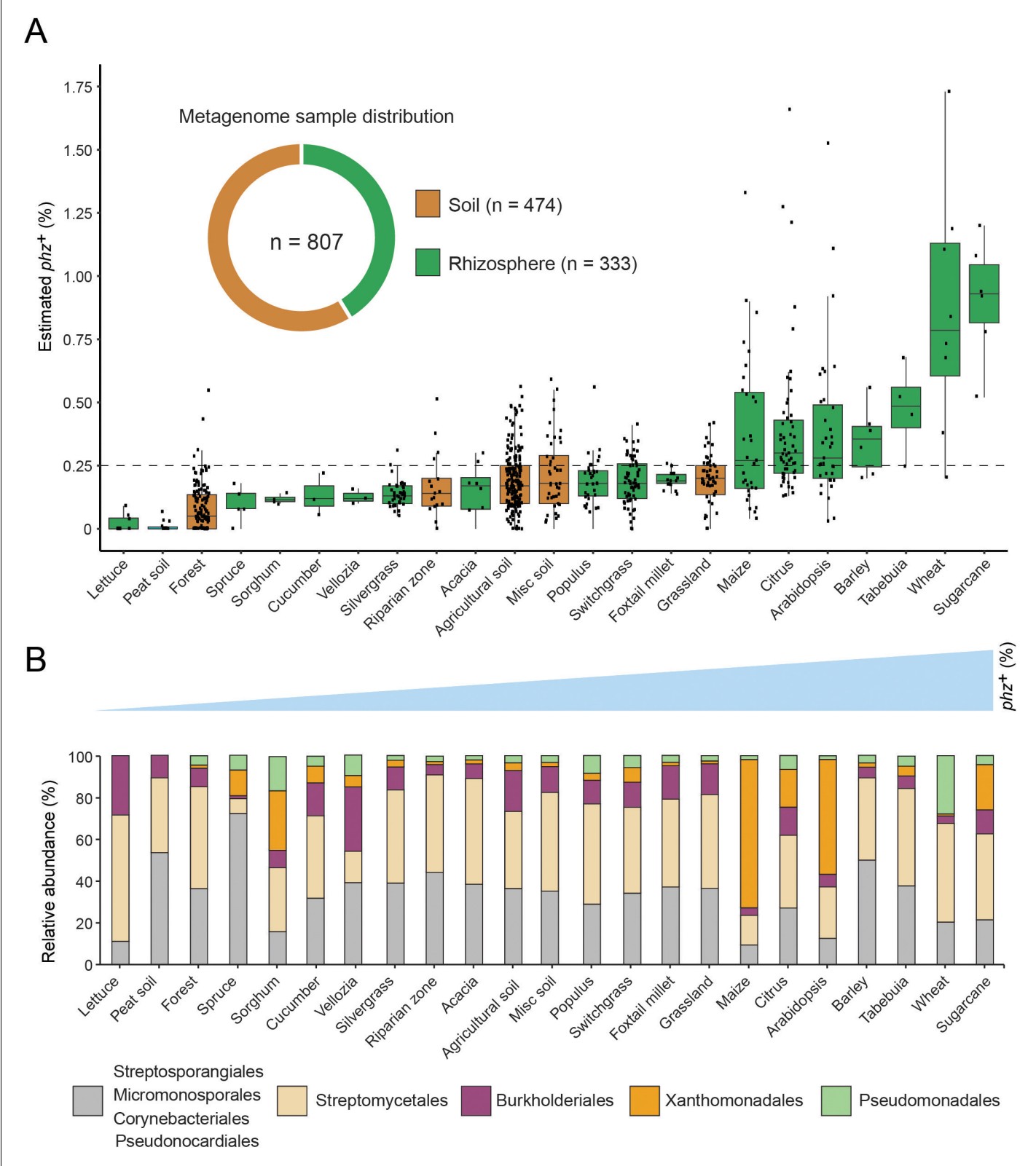

**Figure 4.** Global analysis of phenazine biosynthesis across diverse soil and plant habitats. (A) Distribution of *phz*[+] bacteria levels per each habitat (percent of all bacteria). Soil and rhizosphere samples are depicted in brown and green, respectively. Each boxplot represents a different habitat as annotated in the x-axis. Boxplot central line shows the median and all samples are shown as black dots. (B) Comparison of the relative abundance of *phz*[+] taxonomic distributions across habitats, considering all individual samples (e.g., the maize bar was calculated using all 32 metagenomes in our

*Figure 4 continued on next page*

*Figure 4 continued*

dataset; Analysis of individual samples can be found in *Figure 4—figure supplement 2*). Order level taxonomic groups are depicted via different colors according to the legend.

The online version of this article includes the following figure supplement(s) for figure 4:

**Figure supplement 1.** Metagenomic sample category distributions.
**Figure supplement 2.** Taxonomic analysis of *phz*⁺ rich metagenomes.
**Figure supplement 3.** Phenazine biodegradation gene abundances per habitat.

---

*phz*⁺ species that commonly appears in both soil and rhizosphere samples (*Figure 3D–E*). In contrast, the phenazine gene signatures of other species are more sample-specific, for example, phenazine genes similar to those of *S. luteus* TRM45540 are abundant in the *A. thaliana* rhizosphere, while those of *Streptomyces* sp. Root1310 were most abundant in sugarcane. Furthermore, we find seven *Streptomyces* species that are >2 fold enriched in plant-associated samples. Intriguingly, the genomes of five of these *Streptomyces* contain the mevalonate pathway for phenazine prenylation (*Figure 3E–F*), implying these particular types of phenazines could be globally enriched in rhizosphere microbiomes.

In addition to *Streptomyces*, low levels of other *phz*⁺ actinobacteria such as Streptosporangiales and Pseudonocardiales are common in many habitats (*Figure 4B*; *Figure 4—figure supplement 2*). In contrast to these ubiquitous, we find that other species occur in a more localized manner. For example, *Pseudomonas* is detected at high levels almost exclusively in our Washington wheat samples and Myxococcales are mainly detected in the barley microbiome (*Figure 4—figure supplement 2*; *Supplementary file 4*). While Burkholderiales are commonly detected, their levels are highest in agricultural field soils (*Figure 4—figure supplement 2*; *Figure 4B*). Surprisingly, we detected Xanthomonadales *phz*⁺ bacteria in 22% (50/229) of our *phz*⁺ rich samples (*Figure 4—figure supplement 2*; *Figure 4B*). High levels of these *phz*⁺ bacteria occur exclusively in plant rhizosphere microbiomes, preferentially colonizing maize, citrus, sugarcane, and *A. thaliana* (*Figure 4B*; Figure S5). In addition to their notable distribution, Xanthomonadales bacteria represent the most abundant species in our data, reaching up to 2.7% of all bacteria in the maize rhizosphere (*Supplementary file 4*).

Our analysis also measures the levels of bacteria capable of phenazine biodegradation (*Figure 4—figure supplement 3*; *Figure 2E*). While we find scarce evidence for the *podA* pyocyanin oxidative demethylation gene (*Costa et al., 2017*) in these soil and plant environments, 21 samples contain substantial levels of *phdA*⁺ bacteria (at least 0.25%; *Supplementary file 4*), whose gene product begins the degradation of PCA (*Costa et al., 2018*). This result is consistent with the enrichment of soil bacteria that produce PCA and other phenazines rather than pyocyanin, which is more commonly associated with human chronic infections (*Sismaet et al., 2016*; *Wilson et al., 1988*). Notably, 71% (15/23) of these samples were annotated as switchgrass rhizospheres, with six samples also containing similar or higher *phz*⁺ bacterial levels composed of mostly *Streptomyces* (*Figure 4—figure supplement 3*). Although rare in our studied environments, these data provide the first indication that *phz*⁺ and phenazine-degrading bacteria can occupy the same environment at similar abundances.

## Characterization of phenazine production in *Dyella japonica*

Our quantitative biogeographic analysis indicates that a group of *phz*⁺ Xanthomonadales frequently colonize the rhizospheres of several major crops, at levels equal to or higher than those of previously studied bioactive pseudomonads, suggesting they could be agriculturally relevant (*Supplementary file 4*). We therefore decided to study these previously unknown and potentially significant plant-microbe associations in more detail.

Our *phz*⁺ reference database contains only two distantly related groups belonging to the Xanthomonadales: *Lysobacter* and *Dyella* (*Figure 1C*; *Supplementary file 1*). Strikingly, we find that our metagenomic sequencing data almost exclusively maps to *Dyella* rather than *Lysobacter* (*Figure 5A*). However, these *phz*⁺ *Dyella* were only recently isolated from the *A. thaliana* rhizosphere and were not studied with respect to their putative phenazine biosynthesis capabilities (*Levy et al., 2018*). In contrast, the phenazine repertoire of *L. antibioticus*, as well as the specific genes responsible for their synthesis and modification, were recently characterized in depth. Specifically, the *L.*

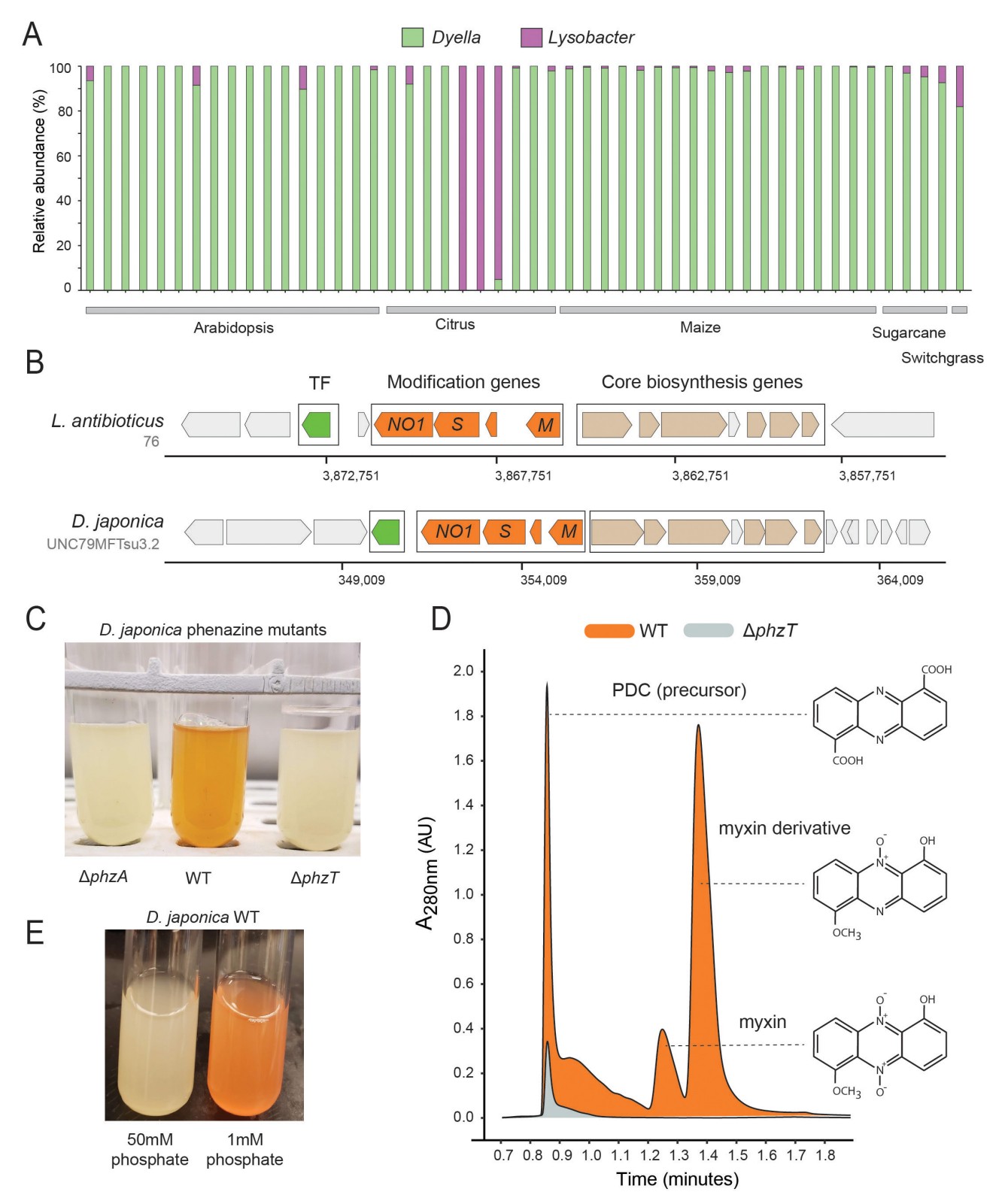

**Figure 5.** Phenazines produced by *D. japonica*. (**A**) Relative abundance of *Lysobacter* and *Dyella* out of all Xanthomonadales in high *phz*[+] samples (**B**) Genomic regions containing the core phenazine biosynthesis genes (tan) and phenazine modification genes (orange) in *L. antibioticus* and *D. japonica*. A conserved transcription factor (TF) is marked in green and all other genes shown in gray. (**C**) *D. japonica* Δ*phzA*, WT and Δ*phzT* culture tubes following 48 hr growth in minimal defined medium. (**D**) WT (orange) and Δ*phzT* (gray) spent-culture media LC-MS results showing UV absorbance at 280

*Figure 5 continued on next page*

*Figure 5 continued*

nm. Detected phenazine structures are shown to the right with a dashed line indicating the position where their mass was detected. (E) WT *D. japonica* cultures grown for 24 hr in minimal defined media containing either 1 mM or 50 mM phosphate.

The online version of this article includes the following figure supplement(s) for figure 5:

**Figure supplement 1.** LC-MS UV-vis profiles in WT and mutant *D. japonica* cultures.

**Figure supplement 2.** LC-MS mass traces of specific phenazines in WT and mutant *D. japonica*.

---

*antibioticus* gene products *La*PhzS and *La*PhzNO1 were shown to catalyze the decarboxylative hydroxylation and N-oxidations of PDC to generate the phenazine iodinin (*Zhao et al., 2016*). In addition, the SAM-dependent O-methyltransferase, *La*PhzM, was later found to convert iodinin into the highly toxic phenazine, myxin (*Jiang et al., 2018*). We compared the biosynthesis clusters of both *L. antibioticus* and *D. japonica* and found similar gene content between these organisms, as well as a conserved operonic structure. Notably, an adjacent putative transcription factor (TF) was also conserved between these organisms. Thus, *D. japonica* is likely to produce phenazines that are similar or identical to those previously detected in *L. antibioticus* (*Figure 5B*).

To test this prediction in *Dyella*, we acquired the *D. japonica* UNC79MFTsu3.2 strain and generated clean deletion mutants in either the phenazine biosynthesis gene, *phzA*, or the conserved transcription factor, *N515DRAFT_3255* (denoted here as *phzT*), present near the operon (*Figure 5B*). Most phenazines are brightly pigmented. Indeed, while WT *D. japonica* produce a bright orange pigment when grown in minimal medium for 48 hr, both mutant cultures remain completely colorless, indicating that they are phenazine deficient (*Figure 5C*). We compared the metabolites found in WT, Δ*phzA* and Δ*phzT* spent culture medium using High-pressure Liquid Chromatography Mass Spectrometry (HPLC-MS) and searched for known phenazine masses and characteristic UV absorbances (*Jiang et al., 2018*; *Myhren et al., 2013*). We detected several phenazines previously described in *L. antibioticus* in the WT cultures but not in the phenazine deficient mutants (*Figure 5D*). These included the precursor PDC, and the potent antimicrobials myxin and a reduced myxin derivative (1-Hydroxy-6-methoxyphenazine-N10-oxide) (*Chowdhury et al., 2012*; *Weigele et al., 1970*; *Figure 5D*; *Figure 5—figure supplements 1* and *2*). These results suggest that the conserved transcription factor deleted in our analysis is an essential regulator of phenazine biosynthesis in this organism.

While the phenazines produced by *D. japonica* have the chemical potency required for pest control, it is important to understand under which conditions these metabolites are preferentially made. Recently, it has been suggested that redox active metabolites such as phenazines might increase phosphate solubility in soils, due to their potential to liberate absorbed phosphate from iron minerals through the process of reductive mineral dissolution (*Dahlstrom et al., 2020*). In *P. aeruginosa*, phosphate limitation has been implicated in the induction of phenazine biosynthesis (*Whooley and McLoughlin, 1982*; similar trends have been seen for other antibiotics made by diverse bacteria [*Martín, 1977*]). Perhaps not surprisingly, therefore, we find similar induction of phenazine biosynthesis in phosphate limited *D. japonica* cultures, even after only 24 hr (*Figure 5E*).

## *D. japonica* robustly colonizes maize seedling roots in a gnotobiotic laboratory model

Our metagenomic analyses above identified *Dyella* in the rhizospheres of multiple crop species with the highest abundance in maize (*Figure 4B*). To validate and characterize *D. japonica* plant colonization patterns in this staple crop, we used a gnotobiotic seedling model (*Niu et al., 2017*). In this setup, maize kernels are first sterilized and then allowed to germinate on water agar under axenic conditions. At 48 hr post germination, the seeds are incubated with *D. japonica* for 30 min and are then planted in sterile ½ MS agar tubes (*Figure 6A*). The seedlings are grown for 7 days and bacterial colonization efficiency is then measured by extracting and plating bacteria from the root-base surface. We find that *D. japonica* accumulates to high levels in this mono-colonization model, reaching an average of $1.5 \times 10^6$ cfu/mg root-base (*Figure 6B*). These colonization levels are comparable to strong maize colonizers tested in this model (*Niu et al., 2017*) and are similar to those previously reported for plant growth-promoting *Pseudomonas fluorescence* in a tomato seedling model (*Simons et al., 1996*). We note that phenazine-deficient mutants accumulate to a slightly higher level

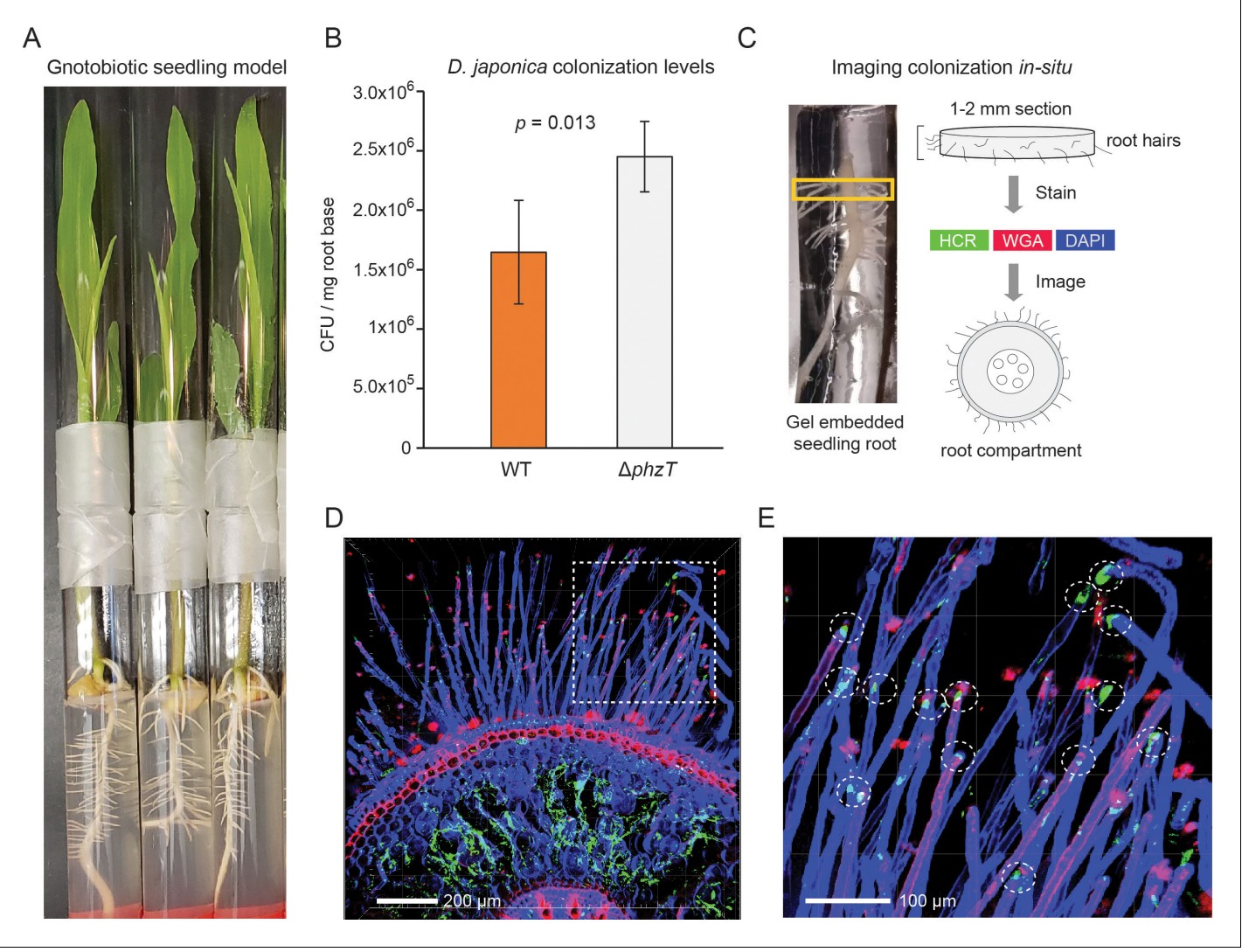

**Figure 6.** *D. japonica* maize seedling colonization. (**A**) Example of 7-day-old maize seedlings grown in an agar-based gnotobiotic system inside two glass tubes connected by parafilm. (**B**) Colonization efficiency of *D. japonica* WT (orange) or Δ*phzT* (gray) in a mono-colonization assay (n = 5 for each genetic background). Error bars represent standard deviation. P-value for a two-tailed t-test comparison is shown in the panel. (**C**) Illustration of the sample preparation process for in situ imaging of bacterial colonization. Roots are encased in a stabilizing hydrogel and root-base sections are extracted and labeled before imaging. (**D**) Maximal-projection image depicting bacteria (green) and plant tissue via DAPI (blue) and WGA (red). (**E**) Zoom-in on the dashed box in panel D. Encircled are root hair tips colonized by bacteria (green patches).

(1.5-fold; p=0.013; t-test) (*Figure 6B*). These results suggest that the *A. thaliana* isolated *D. japonica* is a robust maize colonizer and that phenazine production in this mono-colonization assay has a slight adverse effect on colonization efficiency.

In addition to viable counts, we also imaged the in planta colonization patterns of *D. japonica* (*Figure 6C–E*). We fixed and permeabilized seedling roots previously colonized with *D. japonica* and then encased them in a clear stabilizing hydrogel (*Figure 6C*). The root base was sectioned to produce 1–2 mm slices that were optically cleared. Bacteria were labeled using the Hybridization Chain Reaction (HCR) fluorescence in situ technique (*Choi et al., 2014*), targeting the 16S rRNA (Methods). In contrast to fluorescence in situ hybridization (FISH), HCR can enable low magnification analysis of large tissues due to its fluorescent signal amplification. Plant tissue was labeled with 4′,6-diamidino-2-phenylindole (DAPI) and rhodamine-conjugated lectin (wheat germ agglutinin [WGA]) stains. The labeled root sections were then imaged using confocal fluorescent microscopy (*Figure 6C–E*). Bacterial HCR signal indicative of microaggregate formation was readily visible within the root inner

compartment is readily detected (*Figure 6D*). In addition, root hair tips appeared to be preferentially colonized by *D. japonica* (*Figure 6E*).

## Discussion

This is the first study to quantitatively chart the biogeographic distribution of phenazine biosynthesis and biodegradation genes in natural and agricultural habitats. We developed and field-validated a computational method to analyze >800 metagenomes from across the world and show that phenazine biosynthesis is a highly prevalent trait that is enriched in plant microbiomes relative to bulk soils. In contrast, known phenazine biodegradation genes are rarely detected in these environments. Our data highlight new and potentially important associations between specific *phz*⁺ bacterial clades and several major crops. We experimentally validate a novel and frequent association between maize and *D. japonica*, identifying bioactive phenazines produced by this bacterium as well as genetic and environmental factors involved in their regulation. Furthermore, we show that *D. japonica* is a robust maize seedling colonizer that occupies the root interior and root hair tips.

While the majority of research on phenazine production has been in pseudomonads (*Chincholkar and Thomashow, 2013*), our study demonstrates that most crops and agricultural soils are colonized by different *phz*⁺ bacteria, the most common of which are *Streptomyces*. Notably, our analysis discovered that *phz*⁺ *Streptomyces* occur even in relatively well-studied environments such as the Inland Pacific Northwest wheat fields, where pseudomonads have been considered to be the main producers. It has long been known that *Streptomyces* bacteria synthesize highly bioactive metabolites and antibiotics, including phenazines (*Turner and Messenger, 1986*), and some have been characterized as important residents of the plant-root microbiome (*Bulgarelli et al., 2012*; *Rey and Dumas, 2017*). Indeed, our data suggest that particular *Streptomyces* species are more common in plant rhizospheres compared with bulk soils. Notably, many of these species carried the mevalonate pathway operon involved in synthesizing prenylated phenazines (e.g., endophenazines) (*Saleh et al., 2009*). The potential roles of prenylated phenazines in bacterial physiology, root colonization and/or pathogen inhibition are completely unknown and represent an exciting research direction. Our data suggest that prenylated phenazines could be common in the *A. thaliana* rhizosphere, presenting an attractive model system for studying these metabolites, in planta. That phenazine degradation genes are generally more scarce than phenazine production genes, with some notable exceptions (*e.g.* switchgrass rhizospheres), suggests that phenazines might be expected to rise in abundance in certain crop soils as the climate warms (*Maestre et al., 2015*; *Mavrodi et al., 2012*); direct measurements of phenazine concentrations in soils will be needed to test this prediction.

The identification of *D. japonica* as the most abundant *phz*⁺ bacteria in our global analysis was our most unexpected and striking result. This recently discovered bacterium was found almost exclusively in plant rhizosphere samples and was highly enriched in citrus, sugarcane and maize crops, where it reached up to 2.7% of the entire microbiome (~7 fold higher than our measured *phz*⁺ *Pseudomonas* in wheat). *D. japonica* is a robust seedling mono-colonizer that appears to accumulate within the root interior as well as the root hair tips, where root exudates levels are high (*Canarini et al., 2019*). Further experiments in soil-grown maize will be needed to further establish the relevance of these colonization patterns. Notably, *D. japonica* can produce the highly toxic phenazine myxin, and upregulates phenazine production when limited for phosphate—an environmental stress likely to become more common as phosphorus fertilizers become scarce (*Cordell et al., 2009*). Intriguingly, a mutant deficient in phenazine biosynthesis exhibited somewhat greater plant colonization than the wild type in this model. Because phenazines can have a fitness cost under some conditions in isogenic cultures (*Meirelles and Newman, 2018*), this result is perhaps unsurprising. Moreover, based on the physiological functions of phenazines (*Chincholkar and Thomashow, 2013*), we would expect any fitness advantage due to phenazine production to manifest at a later stage of growth, be the phenazine producer in mono-culture or in the presence of competing organisms (*LeTourneau et al., 2018*; *Mazzola et al., 1992*). The natural abundance of *phz*⁺ *D. japonica* in the microbiomes of economically important crops, as well as its bioactive phenazines suggest that this organism could have important agricultural effects.

In this work, we developed and field-validated a simple process for measuring the abundance of phenazine producing and biodegrading species. The basic premise of this approach is that a

measure for total-bacterial signal can be calculated using a defined set of ubiquitous, single-copy genes (*Manor and Borenstein, 2015*; *Parks et al., 2018*). This reference signal can then be used to normalize gene abundances between samples in a robust manner (*Manor and Borenstein, 2015*). We use the fact that phenazine biosynthesis genes almost always appear in a single genomic copy to interpret our data as the abundance of phenazine producers out of all bacteria in a given environment (percent producers). We note that this more ecologically intuitive interpretation is only possible when the genomic copy of a given gene can be accurately inferred. Another potential limitation of this general approach is its reliance on a database of gene sequences that might not capture all environmental sequence variants. This issue can be addressed by defining large and diverse reference sets and/or by setting looser read mapping thresholds. Potential false positive signals can arise if highly similar, yet functionally distinct genes occur in the analyzed environment. This issue is mitigated when studying biosynthesis pathways containing multiple genes, as spurious signals are more likely to be filtered. Alternatively, one could first identify protein regions that are more specific, effectively examining a combination of smaller gene marker sections instead of the whole (*Kaminski et al., 2015*). While we chose to use 25 reference genes in our analysis, the high similarity in coverage between these genes within individual samples (*Figure 2—figure supplement 2*) suggests that a smaller set might be used with similar efficacy. While our approach measures the abundance of specific microbes via their DNA signatures, additional transcriptomic and/or proteomic data would be needed to measure the fraction of these bacteria that is metabolically active. Nonetheless, this approach can readily be extended to other metabolite biosynthesis pathways and functional traits of interest.

In summary, it is now well appreciated that metabolite-mediated interactions play important roles in shaping microbial communities in plant and animals. Yet our ability to leverage this knowledge in ecological settings has been limited by our ignorance of which organisms make particular metabolites in particular niches. As shown here, a culture-independent metagenomic approach can be used to spotlight potential players and traits of interest, allowing for the targeted cultivation of ecologically relevant model organisms. These discoveries open the door for basic research and downstream applications harnessing these microbes for agricultural and medical applications.

# Materials and methods

## Key resources table

| Reagent type (species) or resource | Designation | Source or reference | Identifiers | Additional information |
|---|---|---|---|---|
| Strain, strain background (*Dyella japonica* UNC79MFTsu3.2) | *D. japonica* WT | *Levy et al., 2018* | Genome IMG ID: 2556921674 | WT |
| Strain, strain background (*Dyella japonica* UNC79MFTsu3.2 ΔphzA) | *D. japonica* ΔphzA | This paper | | *phzA* deletion mutant |
| Strain, strain background (*Dyella japonica* UNC79MFTsu3.2 ΔphzT) | *D. japonica* ΔphzT | This paper | | *phzT* deletion mutant |
| Recombinant DNA reagent | pMQ30 | *Shanks et al., 2006* | GenBank: DQ230317.1 | Plasmid for generating deletions in *D. japonica* |
| Genetic reagent (*Escherichia coli* S17-1) | *E. coli* S17-1 | *Simons et al., 1996* | ATCC 47055 | Mobilizing strain that carries the pMQ30 plasmid |
| Software, algorithm | DIAMOND v0.9.22.123 | *Buchfink et al., 2015* | DIAMOND | Mapping translated reads to references |
| Software, algorithm | Trimmomatic v0.38 | *Bolger et al., 2014* | Trimmomatic | Read quality control |
| Software, algorithm | fastq-dump v2.9.2 | NCBI | fastq-dump | Retrieve public metagenomes from SRA |

## Gene reference set construction

A database of phenazine core biosynthesis genes *phzA/B*, *phzD*, *phzE*, *phzF* and *phzG* was constructed by mining IMG-ABC (*Hadjithomas et al., 2015*). The genomic regions predicted by IMG-ABC to contain phenazine biosynthesis genes were filtered to only select regions that contain the majority of required biosynthesis genes (up to one missing gene). The resulting set of organisms and specific genes was used for additional blast-based homology searches in IMG. We limited the number of strain level variants to 10. The final set of 181 bacterial species along with the IMG gene identity numbers used in the study is available in *Supplementary file 1*. The 'total-bacteria' reference set was constructed using the published bac120 gene set (*Parks et al., 2018*). The database was downloaded from the Genome Taxonomy Database (GTDB) https://gtdb.ecogenomic.org/. We selected the top 25 most ubiquitous genes (*Supplementary file 2*) and extracted their protein sequences. The *podA* and *phdA* phenazine biodegradation reference set was constructed using homology searches. The phenazine, biodegradation and total-bacteria sequences were merged and formatted into a DIAMOND database (*Buchfink et al., 2015*). The concatenated *phzA/BDEFG* sequences (for 176 species containing all core genes; *Supplementary file 1*) were used to construct the phenazine phylogenetic tree. The sequences were aligned using MUSCLE (*Madeira et al., 2019*) and the tree was constructed using FastTree 2.1 (*Price et al., 2010*). The tree was drawn with iTOL (*Letunic and Bork, 2019*).

## Metagenome mapping and estimation of $phz^+$ bacteria levels

Single-end Illumina sequencing reads or simulated metagenomes (described below) were mapped to the protein database described above using DIAMOND with the -k 1 option, which outputs the best hit (*Buchfink et al., 2015*). For reads that mapped equally well to more than one target genome the first occurrence in the database was selected by the DIAMOND -k 1 option. For analyses evaluating the abundance of specific species, the analysis was repeated for a subset of samples, and the reads were randomly distributed between all targets. No significant quantitative differences were observed when using the DIAMOND -k 1 option (first occurrence) or randomly distributing the reads between all targets. The data in the figures shows the results of using the DIANMOND -k 1 option. Reads were first trimmed using Trimmomatic (*Bolger et al., 2014*) and trimmed fastq files were converted to fasta and mapped to the reference databases. Translated reads that mapped with less than 80% amino-acid identity were discarded and total read counts per gene were normalized by the hit average gene size. The median normalized scores per category was calculated using either phenazine genes, biodegradation genes, or the reference 'total-bacteria' genes, producing a single score for each set. The frequency of phenazine producers and biodegraders out of all bacteria in the population was estimated by dividing the phenazine score (or biodegradation score) by the reference 'total-bacteria' score.

## Benchmarking using simulated metagenomes

A set of 28 phylogenetically diverse bacteria were selected (*Supplementary file 3*), several of which are highly similar taxonomically to the tested phenazine producers (e.g., *Pseudomonas* or *Streptomyces* species that do not contain the phenazine operon). We tested the accuracy of our approach by generating synthetic metagenomes where a specific phenazine producer (either *P. chlororaphis* 189 or *Streptomyces* sp. NRRL S-646) was spiked in at a known abundance (0, 0.1, 0.25, 0.5, 1, 1.5, 2.5, 3.5 and 5%). Mixed $phz^+$ synthetic metagenomes were constructed by first setting the total $phz^+$ level and then randomly selecting the portion of each $phz^+$ species (e.g., total $phz^+$=5% and *P. chlororaphis* = 1% while *Streptomyces* spp. = 4%). In addition to varying the $phz^+$ bacteria level we also varied the library coverage, sampling a total of 5, 10, 15 or 20M 150nt fragments from each condition (*Figure 2—figure supplement 1*). Each experiment (e.g., $phz^+$ level = 0.25% at 10M reads) was repeated 12 independent times. While the producer level was always set per condition, the relative abundance of each of the remaining 28 bacteria was randomly set in every replicate to improve result robustness. While the frequencies of phenazine producers were set as describe above, each community contained a randomly generated frequency of phenazine degraders. For each metagenome, we selected the appropriate number of reads for each species by randomly selecting 150nt fragments from their assembled genomes. These simulated metagenomes were then analyzed as described above to estimate the $phz^+$ levels.

## Lind site wheat rhizosphere and bulk soil sampling and processing

The Washington State University Lind Dryland Research Station (46.973°N, 118.616°W, 423.7 m above sea level) was sampled on May 6th, 2019 for 6-week-old wheat seedlings (*Triticum aestivum* L. cv. Louise) as well as bulk field soil collected between planted rows. Rhizosphere communities were extracted by shaking plant roots until only 1–2 mm tightly adhering soil was left covering the root system. A single plant was then placed in 20 ml sterile filtered water and vortexed (1 min) and then treated in an ultrasonic bath (1 min) to dislodge root associated bacteria. Indigenous root-associated *phz*+ pseudomonads were enumerated by the modified terminal dilution endpoint assay as previously described (*Mavrodi et al., 2018*). DNA was extracted from rhizospheres and from bulk soil samples using the DNeasy PowerSoil Kit (Qiagen, 12888–100) according to the manufacturer's instructions. Illumina metagenomic DNA libraries were prepared and sequenced at the Millard and Muriel Jacobs Genetics and Genomics Laboratory at the California Institute of Technology.

## Analysis of publicly available shotgun-metagenomes

The Sequence Read Archive (SRA) was mined for shotgun-metagenomic samples collected from soils and plant-roots. The results were filtered to remove poorly annotated and shallow sequenced samples. A total of 799 individual samples were selected for further analysis and are summarized in *Supplementary file 4* along with producer levels and taxonomies. Samples were coarsely classified as either soil (non-plant associated) or rhizosphere (plant-associated) according to sample metadata. Samples were further sub-classified according to the environment from which they were extracted. The Raw fastq files were download using the SRA Toolkit (fastq-dump) and were processed and analyzed as described above, analyzing read1 files only in cases where paired-end data was available.

## Bacterial strains, plasmids, primers, and growth conditions

*D. japonica* UNC79MFTsu3.2 was grown aerobically with shaking at 250 rpm in lysogeny broth (LB) (Difco) or on LB agar plates at 30°C for all cloning and strain construction purposes unless otherwise noted. For phenazine pigment production assays, *D. japonica* was grown overnight in LB, washed twice with sterile water and diluted 1:200 into fresh defined medium containing $KH_2PO_4$ (1.93 g), $K_2HPO_4$ (6.24 g), NaCl (2.5 g), $MgSO_4$ (0.12 g), $NH_4Cl$ (0.5 g), Glucose (20 mM) and amino acids (1x final concentration; Sigma M5550), trace metals and micro-nutrients. For phosphate limitation experiments, the level of phosphate was diluted 50-fold (50 mM to 1 mM) and the medium was buffered with 4-Morpholinepropanesulfonic acid (MOPS; 50 mM; pH 7.0) instead. The cultures were then grown at 30°C for 24–48 hr until pigment production saturated.

## Deletion mutant generation

Unmarked deletions in *D. japonica* were made using a variation of a previously described protocol for *P. aeruginosa*. Briefly, ~1 kb fragments immediately upstream and downstream of the target locus were amplified by PCR (Kappa Hi-Fi) and joined with the vector pMQ30 (*Shanks et al., 2006*) (cut with *Sma*I restriction enzyme) using Gibson assembly (NEB). Primer sequences can be found in *Supplementary file 5*. The assembled construct was transformed into *E. coli* S17-1, and transformants were plated and selected on LB plus gentamicin (20 µg/ml). For all plasmids, a correctly assembled construct was identified by colony PCR and Sanger sequencing (Laragen). Conjugation between *E. coli* S17-1 and the *D. japonica* WT was performed overnight at 30°C on LB plates. *D. japonica* merodiploids containing the chromosomally integrated construct were selected on LB plus gentamicin (20 µg/ml) and chloramphenicol (6 µg/ml) to exclude the donor *E. coli*. The merodiploids were then grown to exponential phase in LB and counter selected on LB agar plates lacking NaCl and containing sucrose (10%, w/v) plus chloramphenicol (6 µg/ml). Deletions were identified by colony PCR and Sanger sequencing.

## LC-MS analysis

*D. japonica* was grown in defined medium as described above until pigment production saturated (24–48 hr) and the spent medium was filtered with 0.2 µm spin filters (Corning #8160), placed in sample vials (Waters #600000668CV) and loaded into the autosampler at 10°C. Samples were run on a Waters LC-MS system (Waters e2695 Separations Module, 2998 PDA Detector, QDA Detector). For each sample, 10 µL was injected onto a reverse phase C-18 column (XBridge #186006035) running a

gradient of 100% $H_2O$ + 0.04% $NH_4OH$ to 70% acetonitrile + 0.04% $NH_4OH$ over 11 min (20 min total run time). UV-Vis and positive MS scans were acquired for each run. The known masses for phenazine previously detected in *L. antibioticus* (*Jiang et al., 2018*; *Zhao et al., 2016*) were identified by comparing among WT, Δ*phzT* and Δ*phzA* cultures. Phenazine UV-absorbance was measured at 254 nm, 280 nm and 288 nm (*Figure 4—figure supplement 2*).

## Gnotobiotic maize colonization assay

Maize seedlings were grown as previously reported (*Niu et al., 2017*) with minor variations. Briefly, *Zea mays* cv. Sugar Buns F1 (se+) (Johnny's Selected Seeds, catalog number: 267) were sterilized for 3 min using 50% commercial bleach with 0.1% Tween and then washed with sterile water four times. The seeds were imbibed in sterile water for 2.5 hr at room temperature and then placed on 1% water agar plates, embryo side up, five per plate, and were germinated in the dark at 25°C for 48 hr. Seedlings were placed in PBS solution containing WT or mutant *D. japonica* at a concentration of $10^6$ cfu/ml and were left to soak for 30 min. The seedlings were then transferred into glass test tubes (16 × 150 mm) containing 20 mL ½ strength Murashige and Skoog Basal Salt Mixture (MS) (Sigma-Aldrich, catalog number: M5524-50L) with 0.75% agar. The seeds were placed in the agar root first and each tube was then closed with a second tube connected using parafilm to allow gas exchange. The plants were grown for 7 days in a 16 hr light (day; 4000 lumens) and 8 hr of dark (night) cycle at 25°C. Colonization levels were calculated by collecting 2-cm-long roots immediately below the seed (root base). Root sections were washed in 1x PBS and any remaining agar pieces were carefully removed using sterile forceps. The roots were then vortexed in tubes containing 1 ml PBS with 3 glass 3 mm beads and then sonicated for 30 s to dislodge root associated bacteria. The solution was serially diluted and plated on LB agar plates and incubated for 48 hr before colony counting. To normalize between roots of different sizes, each root was patted dry and weighed and the total number of cultured bacteria was adjusted per mg root.

## Imaging *D. japonica* colonization in planta

Seedling root systems (7 day old) colonized with *D. japonica* were fixed in a 4% paraformaldehyde 1x PBS solution overnight at 4°C and then washed 3 times using 1x PBS. Samples were then incubated overnight in 4% (vol/vol) 29:1 acrylamide:bis-acrylamide and 0.25% (wt/vol) VA-044 hardener in 1x PBS. The following day, the root samples were moved into an anaerobic hood and left open, but covered, for 5 min to remove headspace oxygen. Capped samples were polymerized for 3 hr at 37°C in a water bath, without shaking. Gel encased roots were sectioned using a scalpel to produce 1–2 mm embedded root samples which were then cleared in a solution of 8% SDS, pH 8.0, at 37°C, for 24 hr. HCR was performed using the EUB338 probe as previously described (*DePas et al., 2016*). The root sections were incubated at RT overnight in PBS 1x plus 50 µg/ml rhodamine-conjugated lectin (wheat germ agglutinin [WGA]), washed in 1x PBS and then incubated for another 24 hr in refractive index matching solution (RIMS) with 1 µg/ml DAPI. Samples were then mounted on slides in 0.9 mm or 1.7 mm CoverWell perfusion chambers (Electron Microscopy Services) with a coverslip on the top. Imaging was performed using a Zeiss LSM 880 confocal microscope with a Plan-Apochromat 10×/0.45-numerical aperture M27 objective. Image analysis was conducted using Imaris imaging software (Bitplane) and the FIJI distribution of ImageJ.

## Data and software availability

The metagenomic DNA-sequencing data generated in this study were deposited in the Sequence Read Archive (SRA) under accession PRJNA634917. All public SRA samples analyzed in this study are indicated in *Supplementary file 4*. Code can be found at: https://github.com/daniedar/phenazines (*Dar, 2020*; copy archived at https://github.com/elifesciences-publications/phenazines).

## Acknowledgements

We thank Jeff Dangl and Adam Deutschbauer for kindly providing us with the *D. japonica* UNC79MFTsu3.2 strain. We also thank Darcy McRose for help with LC-MS sample preparation and analysis, Megan Bergkessel and Kurt Dahlstrom for their help with mutant generation, Will DePas for assistance with HCR, as well as the rest of the Newman lab for fruitful discussions and comments. We also thank Gil Sharon and Alon Philosof for critically reading the manuscript. We thank Mingming

Yang with assistance in collecting, processing, and performing viable counts of phenazine-producing pseudomonads from Washington State University's Lind Dryland Research Station. We also thank Bruce Sauer and Brian Fode for plot maintenance. Grants to DKN from the NIH (1R01AI127850-01A1) and ARO (W911NF-17-1-0024) supported this work. DD was supported by the Rothschild foundation, EMBO Long-Term, and the Helen Hay Whitney postdoctoral fellowships, as well as a Geobiology Postdoctoral Fellowship from the Division of Geological and Planetary Sciences, Caltech.

## Additional information

### Funding

| Funder | Grant reference number | Author |
| --- | --- | --- |
| Helen Hay Whitney Foundation | | Daniel Dar |
| Army Research Office | W911NF-17-1-0024 | Dianne K Newman |
| National Institutes of Health | 1R01AI127850-01A1 | Dianne K Newman |
| Rothschild Foundation | | Daniel Dar |
| California Institute of Technology | GPS Division Geobiology Postdoctoral Fellowship | Daniel Dar |
| EMBO | Long-Term Postdoctoral Fellowship | Daniel Dar |

The funders had no role in study design, data collection and interpretation, or the decision to submit the work for publication.

### Author contributions

Daniel Dar, Conceptualization, Data curation, Formal analysis, Funding acquisition, Validation, Investigation, Visualization, Methodology, Writing - original draft, Writing - review and editing; Linda S Thomashow, David M Weller, Investigation, Writing - review and editing; Dianne K Newman, Conceptualization, Supervision, Funding acquisition, Writing - original draft, Writing - review and editing

### Author ORCIDs

Daniel Dar (iD) https://orcid.org/0000-0002-6650-5488
Dianne K Newman (iD) https://orcid.org/0000-0003-1647-1918

### Decision letter and Author response

Decision letter https://doi.org/10.7554/eLife.59726.sa1
Author response https://doi.org/10.7554/eLife.59726.sa2

## Additional files

### Supplementary files

• Supplementary file 1. List of phenazine producer genomes and gene accessions used in study.

• Supplementary file 2. List of reference universal single-copy genes used in study.

• Supplementary file 3. List of species and genome accessions used to construct simulated metagenomes.

• Supplementary file 4. SRA sample information and analysis results. Phenazine producer and degrader levels, along with relative abundance of specific taxonomic groups is included for each analyzed sample.

• Supplementary file 5. List of the oligos used in constructing the *D. japonica* mutant strains.

• Transparent reporting form

## Data availability

The metagenomic DNA-sequencing data generated in this study were deposited in the Sequence Read Archive (SRA) under accession PRJNA634917. All public SRA samples analyzed in this study are indicated in table S4. Code can be found at: https://github.com/daniedar/phenazines (copy archived at https://github.com/elifesciences-publications/phenazines).

The following dataset was generated:

| Author(s) | Year | Dataset title | Dataset URL | Database and Identifier |
|---|---|---|---|---|
| Dar D, Newman DK | 2020 | Phenazine producers in the wheat rhizosphere | https://www.ncbi.nlm.nih.gov/bioproject/PRJNA634917/ | NCBI BioProject, PRJNA634917 |

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
