## [Decision Letter]

**Acceptance summary:**

This work offers an innovative approach that combines computational analysis with experimental studies to explore the production of the antibiotic phenazine by bacteria in natural environments. In addition to detecting microorganisms potentially useful in agricultural applications, this approach can also be extended to study other ecologically relevant microbial genes and properties.

**Decision letter after peer review:**

Thank you for submitting your article "Global landscape of phenazine biosynthesis reveals species-specific colonization patterns in soils and crop microbiomes" for consideration by *eLife*. Your article has been reviewed by three peer reviewers, and the evaluation has been overseen by a Reviewing Editor and Gisela Storz as the Senior Editor. The reviewers have opted to remain anonymous.

The reviewers have discussed the reviews with one another and the Reviewing Editor has drafted this decision to help you prepare a revised submission.

Summary:

Phenazines are natural antibiotics produced by bacteria, found mainly in soils and in the plant rhizosphere, that can also have beneficial effects on crops. However, the complexity of the plant-associated microbiome and the difficulty to culture microorganisms frustrate the identification of phenazine producers that might be relevant to crop health. In this elegant and well-designed work, Dar et al. explore the potential of phenazine production in natural environments using an innovative shotgun-metagenomics approach that is then validated experimentally. They first devised a computational procedure to measure phenazine producing bacteria, which was validated in silico and then used to provide a global view of phenazine biosynthesis and degradation by analyzing >800 soil and plant-associated metagenomes. Based on their biogeographical metagenomic analysis they identified an abundant yet less studied phenazine-producing bacterium and proceeded to characterize its interaction with a plant host of agricultural importance. This study not only increases our appreciation of phenazine biosynthesis and biodegradation in the environment but also provides novel tools for exploring specific genes and plant-microbe associations of agricultural potential.

Essential revisions:

1) The authors make claims about horizontal gene transfer of phenazine biosynthetic operons between phyla. While these seem well-supported overall, there are a number of things unclear about the phylogenetic analysis:

a) which tree reconstruction algorithm was used? The paper refers to MUSCLE, but as far as I know this is a multiple-sequence alignment algorithm, not an algorithm for phylogenetic tree reconstruction.

b) How strong is the bootstrap support for the main branches in the tree?

c) The authors use a concatenated alignment. Have the authors checked whether the individual gene trees are congruent, to exclude e.g. recombination of phenazine operons during evolution?

2) Reference databases are used to map metagenomic reads to and thus assess the frequency of *phz* producers across the bacterial fraction of microbial communities. The overall approach appears sound, although I wonder how it deals with ambiguous mapping (reads that can map equally well to two or more reference sequences), as this is not explicitly explained in the paper. When mentioning that 'reads that mapped with less than 80% identity', it is also not clear whether this indicates amino acid sequence identity or nucleotide sequence identity. Also, how was this threshold determined to be optimal in terms of specificity vs. sensitivity? Finally, are there any clades of uncultivated microbes with phenazine biosynthetic capabilities that the approach could be missing?

3) For the computational analysis, why were 25 genes used from the ubiquitous single copy gene set? Does the estimate of 'total bacteria' change if you do 10 or 40? Do differences in average genome lengths alter this calculation, and is this something you need to worry about? How does your work compare to other approaches assessing gene abundances per bacteria in metagenomic samples, such as MicrobeCensus?

4) The abundance statistic described in Figure 2A normalizes *phz*^+^ genes to total-bacterial score or Reads Per Kilobase levels, a method analogous to the widely used RPKM metric used to measure gene expression which has certain pitfalls (https://rnajournal.cshlp.org/content/early/2020/04/13/rna.074922.120.abstract). The assumption that the RPKM or here, RPK, values are normalized and therefore comparable across multiple samples can be misleading as RPK represents the abundance of the DNA fragment from a population of sequenced DNA fragments. This population represents distinct composition and titer of microbiota that can be unique to different sets of samples (e.g. citrus or maize rhizosphere). I would like the authors to clarify how the RPK statistic proposed here addresses this issue.

5) The validation using simulated metagenomes is set up well but could be explained in more detail. To more clearly show that the approach is able to accurately map frequencies of phenazine operons that are not themselves present in the *phz* reference database the authors may consider doing a cross-validation analysis by leaving the sequences that are spiked into the simulated metagenome out of the reference dataset. This would allow quantification of how similar a sequence needs to be to one of the reference sequences to still be accurately identified with the mapping procedure. In 2B why is the relationship not linear between estimated and known *phz*^+^?

6) For the field validation, can you show a 'predicted' vs. 'experimental' *phz* plot? You show estimates in 3A, how do these relate to your culture-based approach? This is a key point of your study and needs a figure to demonstrate the validation.

7) Since the vast majority of the metagenome sample are from public data, did the authors consider the impact of DNA quality and variability in sample biomass (https://pubmed.ncbi.nlm.nih.gov/25329041/)? Were any of the public samples from a longitudinal study where the impact of environmental conditions on the distribution of *phz*^+^ genes could be explored?

8) Although most of the reads mapped to representative genomes for pseudomonads and *Streptomyces*, did the authors find any off-target mapping for translated DNA reads to proteins from other gene families? Short reads can represent domains shared by *phi*^+^ genes and other gene families and give rise to cross-mapping and over or underestimation. This could possibly bias the mapping counts for *phi*^+^ genes.

9) What is the intuition for why phosphate limitation turns on phenazine production?

10) Given that the authors did not include any transcriptomics experiments, there is no direct indication of what proportion of the microbes represented by the sequenced DNA fragments are transcriptionally and metabolically active in each sample. This limitation should be mentioned in the manuscript.

---

## [Author Response]

Essential revisions:1) The authors make claims about horizontal gene transfer of phenazine biosynthetic operons between phyla. While these seem well-supported overall, there are a number of things unclear about the phylogenetic analysis:a) Which tree reconstruction algorithm was used? The paper refers to MUSCLE, but as far as I know this is a multiple-sequence alignment algorithm, not an algorithm for phylogenetic tree reconstruction.

We used the MUSCLE algorithm to perform the MSA using the EMBL-EBI website. This server also provided a phylogenetic tree along with the MSA, which we included in Figure 1C.

b) How strong is the bootstrap support for the main branches in the tree?

Since the original tree did not contain bootstrap values, we redid the tree reconstruction with FastTree, which infers approximately-maximum-likelihood phylogenetic trees (Price, Dehal and Arkin, 2010). The new tree is highly similar to the original tree and indeed, the bootstrap values strongly support the original conclusions. Figure 1C was revised and now contains the new tree along with the bootstrap values.

c) The authors use a concatenated alignment. Have the authors checked whether the individual gene trees are congruent, to exclude e.g. recombination of phenazine operons during evolution?

We did not previously check this possibility. We repeated the MSA and tree construction procedure described above for the *phzF* biosynthesis gene and found that this tree is congruent with the concatenated tree.

2) Reference databases are used to map metagenomic reads to and thus assess the frequency of phz producers across the bacterial fraction of microbial communities. The overall approach appears sound, although I wonder how it deals with ambiguous mapping (reads that can map equally well to two or more reference sequences), as this is not explicitly explained in the paper. When mentioning that 'reads that mapped with less than 80% identity', it is also not clear whether this indicates amino acid sequence identity or nucleotide sequence identity. Also, how was this threshold determined to be optimal in terms of specificity vs. sensitivity? Finally, are there any clades of uncultivated microbes with phenazine biosynthetic capabilities that the approach could be missing?

We thank the reviewers for raising this important point. Ambiguously mapped reads can be dealt with in several ways. In our analysis, using only uniquely mapped reads does not work as we must account for all reads when estimating the total abundance of *phz*^+^ bacteria. We therefore aimed to randomly distribute the reads between ambiguous targets, which would not affect the relatively low-resolution phylogenetic analysis conducted in this work.

However, the reviewer’s comment encouraged us to revisit the “randomness” of our ambiguous target selection in more detail. We found that the DIAMOND top hit provided by the -k 1 option that we used is biased toward sequences that are recorded earlier in the reference database. This has no effect on the total abundance measurements and Order level phylogenetic analysis (as there is essentially no ambiguous mapping between organisms of different Orders). We revisited the higher resolution analyses, namely those in the wheat rhizosphere and *Dyella* vs. *Lysobacter* comparisons. We remapped a subset of the data with DIAMOND, but this time kept all of the hits. We reanalyzed the mapping results after randomly distributing ambiguously mapped reads across all targets. We found that this does not have any significant quantitative effect on any of the conclusions presented in the original manuscript and so kept the original reported results. The only main difference we could find was the specific *Pseudomonas* strains that were commonly represented in the rhizosphere slightly differed as reads were now also mapped to other similar strains.

We have revised the Materials and methods section to explain this in detail as well as the relevant Results section text:

Revised Materials and methods text:

“Single-end Illumina … were mapped to the protein database described above using DIAMOND with the -k 1 option, which outputs a single best hit (Buchfink et al., 2015). […] The data in the figures shows the results of using the DIANMOND -k 1 option”.

And in the relevant Results text:

“Notably, three species in our reference set comprise >87% of the *phz*^+^*Pseudomonas* rhizosphere sequencing reads (Figure 3C). These reads mapped to multiple strains, some of which were more highly represented, including, *P. fluorescens* LBUM223, *P. chlororaphis aureofaciens* 30-84 and *P. orientalis* DSM 17489.”

Regarding, the 80% mapping threshold, we used amino-acids and have now made this clear in every relevant part of the manuscript text. The 80% threshold was based on testing different options during our simulations. To address this comment as well as comment number 5, we have now also conducted a cross-validation analysis that supports the threshold choice. The new data can be seen in the revised Figure 2—figure supplement 1.

Indeed, there is a reasonable possibility that some uncultivated *phz*^+^ clades are missing from our reference set. We point to this limitation in the Discussion section:

“Another potential limitation of this general approach is its reliance on a database of gene sequences that might not capture all environmental sequence variants. This issue can be addressed by defining large and diverse reference sets and/or by setting looser read mapping thresholds.”

3) For the computational analysis, why were 25 genes used from the ubiquitous single copy gene set? Does the estimate of 'total bacteria' change if you do 10 or 40? Do differences in average genome lengths alter this calculation, and is this something you need to worry about? How does your work compare to other approaches assessing gene abundances per bacteria in metagenomic samples, such as MicrobeCensus?

We originally tested simulated data with the top 8 bac120 genes and got relatively similar results to our work with the top 25. However, we chose to increase the number of reference genes to gain more confidence when analyzing environmental metagenomes. As the analysis of ~800 shotgun-metagenomes was a computationally intensive task, we opted to use 25 rather than the entire 120 (there are ~25,000 sequences for each gene).

In retrospect, we believe that similar results would be obtained with ~10 genes. This is supported by the highly similar read coverage across the 25 genes within individual samples. The average coefficient of variation (CV) calculated for these genes is only ~21% (calculated using all ~800 samples). Since similar coverage is measured for these different genes it is quite possible that a more compact set could be used.

We have added a new supplementary figure (Figure 2—figure supplement 2) showing the CV distribution across all analyzed samples and now address this point in the Discussion section to enable others interested in adapting this approach to consider the possibility of using a more efficient reference set.

The revised Discussion now reads:

“While we chose to use 25 reference genes in our analysis, the high similarity in coverage between these genes within individual samples (Figure 2—figure supplement 2) suggests that a smaller set might be used with similar efficacy.”

MicrobeCensus uses a set of universal genes to estimate the average genome size in a metagenome and this metric can correct sampling biases between metagenomic samples. This is important when conducting differential gene abundances using read count values (RPK/RPKM).

In our analysis, we do not compare read sampling values between metagenomes but rather compare genes within the same community (same average genome size). From the MicrobeCensus paper: “The probability of sampling a gene from a community is inversely proportional to the average genome size of that community”. Thus, our metric should be unaffected by the average genome size bias, as far as we understand, as the genes in the same sample are sampled from the same community.

In support of this, our simulations, which suggest the approach is reasonably accurate, contain organisms with variable genome sizes. As the frequencies of individual community members are randomly generated during each simulation, the average genome size also fluctuates. Thus, we believe if our results are somehow affected by a size-dependent bias, the magnitude of this effect in our case is likely to be small.

4) The abundance statistic described in Figure 2A normalizes phz^+^ genes to total-bacterial score or Reads Per Kilobase levels, a method analogous to the widely used RPKM metric used to measure gene expression which has certain pitfalls (https://rnajournal.cshlp.org/content/early/2020/04/13/rna.074922.120.abstract). The assumption that the RPKM or here, RPK, values are normalized and therefore comparable across multiple samples can be misleading as RPK represents the abundance of the DNA fragment from a population of sequenced DNA fragments. This population represents distinct composition and titer of microbiota that can be unique to different sets of samples (e.g. citrus or maize rhizosphere). I would like the authors to clarify how the RPK statistic proposed here addresses this issue.

The reviewer is correct to point that using RPK and RPKM values when comparing between samples is not trivial and should be done with care. Indeed, these issues, among others are what originally encouraged us to devise a different way of comparing between samples.

In our analysis, we calculate the RPK values for phenazine and reference genes. However, we only compare these within the same sample, which is used to extract the percent *phz*^+^ bacteria. This approach is robust to changes in effective coverage, which can be caused by different titers of microbiota relative to host DNA or other microbes.

To make this clearer we revised the relevant Results text, which now reads:

“While metagenomic sequencing reads can … One major complication is the uncertainty regarding the composition of the background bacterial community as well as other confounding factors that vary among environments. […] This variability in “effective coverage” confounds normalization by library coverage and thus limits comparison between samples.”

5) The validation using simulated metagenomes is set up well but could be explained in more detail. To more clearly show that the approach is able to accurately map frequencies of phenazine operons that are not themselves present in the phz reference database the authors may consider doing a cross-validation analysis by leaving the sequences that are spiked into the simulated metagenome out of the reference dataset. This would allow quantification of how similar a sequence needs to be to one of the reference sequences to still be accurately identified with the mapping procedure. In 2B why is the relationship not linear between estimated and known phz^+^?

We have added more detail and explicit examples in the Results and Materials and methods section describing the simulations. In addition, we conducted the cross-validation analysis suggested by the reviewers by constructing a new database that is without the spiked-in producer genomes. We remapped the same simulated data to this database. We then analyzed the accuracy of different thresholds (100, 95, 90, 85, 80) in estimating the levels of the missing organism. The results strongly support our choice of an 80% amino-acid identity threshold and show that it is reasonably balanced in terms of its specificity and sensitivity.

These new data are now shown in the revised Figure 2—figure supplement 1.

In Figure 2B, the relationship is visually a little confusing as the boxplot x-axis is not exactly proportional to the y-axis as it is categorical. The linear relationship can be seen in the high Pearson correlation value reported, and in the Figure 2E scatter plot.

6) For the field validation, can you show a 'predicted' vs. 'experimental' phz plot? You show estimates in 3A, how do these relate to your culture-based approach? This is a key point of your study and needs a figure to demonstrate the validation.

Indeed, the comparison between our new culture-independent approach and the standard culture-dependent method represents an important part of this manuscript.

In Figure 3A, we show the percentage of all *phz*^+^ bacteria in the rhizosphere, as measured using our metagenomic approach and in 3B, we find that on average, only about half of these *phz*^+^ bacteria are pseudomonads. The percentage calculation is only possible in the culture-independent analysis, which considers the total bacterial DNA in the environment via the universal reference genes. In contrast, the culture-dependent data specifically enumerates the *phz*^+^ pseudomonads. While counting the total cultured bacteria can be attempted, the uncultured bacterial fraction, which is often quite substantial and culturing condition-dependent, is unaccounted for. Thus, each approach provides a useful piece of information that is not directly comparable. We therefore opted to use a different approach. Instead of a direct comparison, we performed a calculation that uses both metrics to see if these can be used to generate a reasonable prediction of the total bacterial counts in this environment.

To make this clearer, we revised the relevant Results text, which now reads:

“While the enrichment-culture and metagenomic approaches are not directly comparable, they can be used together to extrapolate the number of bacteria in the rhizosphere. Indeed, combining the average *phz*^+^ pseudomonad cfu/g with their average estimated percent out of all bacteria suggests a wheat seedling rhizosphere bacterial load of ~10^9^ bacteria per gram roots, in agreement with other rhizosphere load estimates (Bakker et al., 2013; Mavrodi et al., 2018).”

7) Since the vast majority of the metagenome sample are from public data, did the authors consider the impact of DNA quality and variability in sample biomass (https://pubmed.ncbi.nlm.nih.gov/25329041/)? Were any of the public samples from a longitudinal study where the impact of environmental conditions on the distribution of phz^+^ genes could be explored?

We did not explicitly consider this possible source of bias in our analysis. While this parameter has the potential to affect community composition, it is also one that is difficult to control and correct for, especially using a large public datasets, in which deposited samples do not generally report the DNA quality or concentration used. While specific samples might have variable DNA quality or levels, we were generally able to analyze multiple (often dozens) of samples per environment. Thus, we believe that this potential source of bias would have an overall minor effect on the conclusions of this work.

A relevant longitudinal study would be interesting to analyze, however, we did not identify an appropriate one in this dataset.

8) Although most of the reads mapped to representative genomes for pseudomonads and Streptomyces, did the authors find any off-target mapping for translated DNA reads to proteins from other gene families? Short reads can represent domains shared by phi^+^ genes and other gene families and give rise to cross-mapping and over or underestimation. This could possibly bias the mapping counts for phi^+^ genes.

Indeed, this is an important point which have considered throughout this work. Early on we found that PhzC had close homologues in many other bacteria and therefore removed it from our analysis to reduce false positives, as described in the Results text:

“During this analysis, we noticed that highly similar copies of the *phzC* gene were common in non-producer genomes and therefore removed this gene from our analysis”.

False negatives, however, are unlikely to occur in our specific case as we are using a limited protein database composed of phenazine genes and reference genes only (off targets are not presented as options).

More generally, our approach using the median coverage of 5 different phenazine biosynthesis genes, mitigates such effects. We present these issues in the Discussion section:

“Potential false positive signals can arise if highly similar, yet functionally distinct genes occur in the analyzed environment. […] Alternatively, one could first identify protein regions that are more specific, effectively examining a combination of smaller gene marker sections instead of the whole (Kaminski et al., 2015)”.

9) What is the intuition for why phosphate limitation turns on phenazine production?

Fair question! We have revised the text to explain why we focused on phosphate as a variable:

“Recently, it has been suggested that redox active metabolites such as phenazines might increase phosphate solubility in soils, due to their potential to liberate absorbed phosphate from iron minerals through the process of reductive mineral dissolution (Dahlstrom et al., 2020). […] Perhaps not surprisingly, therefore, we find similar induction of phenazine biosynthesis in phosphate limited *D.japonica* cultures, even after only 24h (Figure 5E).”

10) Given that the authors did not include any transcriptomics experiments, there is no direct indication of what proportion of the microbes represented by the sequenced DNA fragments are transcriptionally and metabolically active in each sample. This limitation should be mentioned in the manuscript.

We have revised the Discussion section to address this point, which now reads:

“While our approach measures the abundance of specific microbes via their DNA signatures, additional transcriptomic and/or proteomic data would be needed to measure the fraction of these bacteria that is metabolically active.”